# Do-Prompt: Causal Interventions Meet Variational Prompt Bottlenecks

Xueting Chen[1]   Jun-Jie Huang[1][†]   Yan Yan[2]   Long Lan[1]   Yuhua Tang[1]   Wenjing Yang[1]

## Abstract

Multi-modal prompt learning is a parameter-efficient approach to adapt large vision–language models to downstream classification tasks. However, prompts can inadvertently evolve into a high-capacity pathway encoding environment-dependent spurious correlations that are only predictive in the source domain, thereby undermining transferability. To address this issue, this paper introduces **Do-Prompt**, a *compress-and-intervene* framework that brings together variational bottlenecks and causal interventions for robust prompt tuning. We model prompts as stochastic latent variables and impose a *variational prompt bottleneck* to explicitly regulate the information transmitted through prompts, effectively mitigating their propensity to memorize spurious nuisance cues. Building on this capacity constraint, we propose lightweight *prompt-level interventions* by perturbing the environment-related prompt components and enforcing prediction consistency under these *do*-style perturbations. This synergistic integration encourages reliance on task-stable, invariant semantics rather than spurious prompt content. Notably, Do-Prompt is plug-and-play compatible with existing multi-modal prompt tuning pipelines with negligible computation overhead. Extensive experiments on base-to-novel generalization, cross-dataset transfer, and ImageNet distribution shifts demonstrate consistent performance gains, with particularly notable improvements on datasets exhibiting pronounced domain or texture biases.

---

[1]College of Computer Science, National University of Defense Technology, Changsha, Hunan, 410073, China [2]Key Laboratory of Multimedia Trusted Perception and Efficient Computing, Ministry of Education of China and Fujian Key Laboratory of Sensing and Computing for Smart City, School of Informatics, Xiamen University, Xiamen, 361005, China. Correspondence to: Jun-Jie Huang <jjhuang@nudt.edu.cn>.

*Proceedings of the 43rd International Conference on Machine Learning*, Seoul, South Korea. PMLR 306, 2026. Copyright 2026 by the author(s).

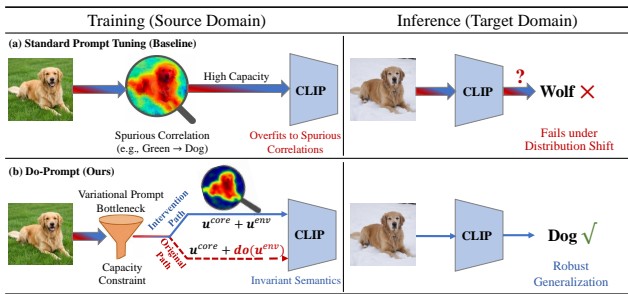

*Figure 1.* **Breaking spurious correlations with Do-Prompt.** (a) Standard prompt tuning may overfit to environment cues that correlate with labels, leading to failures under distribution shift. (b) Do-Prompt constrains prompt capacity with a variational bottleneck and intervenes on the environment part ($\boldsymbol{u}^{\text{env}}$), encouraging predictions to rely on task-stable semantics ($\boldsymbol{u}^{\text{core}}$).

## 1. Introduction

Vision–Language Models (VLMs) emerge as a strong foundation for visual recognition by aligning images and text through large-scale pre-training (Radford et al., 2021). For downstream classification tasks, prompt learning offers an efficient alternative to full-scale fine-tuning by optimizing only a small set of prompt tokens with a frozen backbone (Zhou et al., 2022a). This idea has progressed from fixed textual prompts to instance-conditioned prompting (Zhou et al., 2022b), and further to multi-modal prompt learning that injects prompts into both the vision and language branches to improve cross-modal interaction and transfer capability (Khattak et al., 2023a; Guo & Gu, 2025). Therefore, prompt tuning is now a widely used recipe for data-efficient adaptation of VLMs.

Despite its strong empirical performance, prompt learning has a robustness issue that becomes more evident as prompts become more expressive. Prompts are not just a small set of parameters; they form a learnable information pathway that prioritizes cues which minimize training loss. In practice, training data often comes from a limited range of environmental conditions, such as specific datasets, capture conditions, visual styles, and label priors. When optimized on such constrained data distributions, prompts may encode spurious correlations that yield strong in-distribution predictive performance but degrade rapidly under distribution shift. This vulnerability is especially pronounced

for instance-conditioned and multi-modal prompt learning. Since prompts can depend on the input and interact across modalities, they can more easily entangle task-relevant semantic information with nuisance factors like background, texture, or visual style (Zhou et al., 2022b; Khattak et al., 2023a). As illustrated in Fig. 1(a), this failure case occurs when a model latches onto an environmental cue spuriously correlated with a label during training, leading to catastrophic performance degradation once that cue is altered.

A natural way to mitigate this robustness limitation is to introduce causal thinking into prompt learning. To achieve stable classification under distribution shift, model predictions should be grounded in causal, environment-invariant factors rather than spurious correlations induced by the training data distribution. Recent work has begun exploring causality-guided prompt learning for VLMs (Gao & Dong, 2025; Yu et al., 2024), but most existing methods still optimize deterministic prompts. In practice, deterministic prompts can act as a high-capacity shortcut pathway, and even when the training objective encourages invariance, prompts may still encode and transmit spurious shortcuts unless their information capacity is explicitly constrained. In parallel, the variational information bottleneck (VIB) principle provides a complementary mechanism by learning representations that are predictive of labels while discarding unnecessary information about the input (Alemi et al., 2017; Du et al., 2020). However, VIB has not been systematically integrated into multi-modal prompt learning in a way that explicitly suppresses shortcut propagation through prompts.

This paper introduces **Do-Prompt**, which integrates causal interventions and variational prompt bottlenecks through a simple compress-and-intervene design to mitigate shortcut reliance in multi-modal prompt learning. As illustrated in Fig. 2, we first model prompts as stochastic latent variables inferred from shallow feature context and regularize them with a Kullback–Leibler (KL) divergence term relative to a simple prior. This regularization explicitly constrains the information capacity of the prompt pathway, reducing its ability to encode spurious correlations. Second, we split the latent prompt into a task-stable part and an environment-related part, and apply lightweight *do*-style perturbations (*e.g.*, via masking, swapping and resampling) exclusively to the environment-related part while a prediction consistency objective ensures model outputs remain stable across these perturbations. These two mechanisms operate synergistically, where the variational bottleneck constrains the prompt's capacity to memorize spurious correlations, and the causal intervention actively discourages reliance on environment-specific prompt content, thereby enforcing the learning of invariant task semantics (Fig. 1(b)). Extensive experiments show that Do-Prompt consistently improves robustness-sensitive metrics, with particularly strong gains on datasets exhibiting pronounced domain or texture bias.

## 2. Related Work

**Prompt Tuning.** Prompting offers a lightweight interface to steer foundation models. Beyond manual design, *prompt learning* automates optimization, originally explored in NLP (Shin et al., 2020; Jiang et al., 2020; Liu et al., 2023) and now standard for VLMs. Methods generally fall into three main categories: (1) *Textual prompt learning* (Zhou et al., 2022a;b; Lu et al., 2022; Zhu et al., 2023; Yao et al., 2023; 2024; Park et al., 2024; Tian et al., 2024; Bulat & Tzimiropoulos, 2023; Du et al., 2024b; Li et al., 2025b; Gao & Dong, 2025) optimizes vectors in the language branch to align class semantics with visual features. (2) *Visual prompt learning* (Jia et al., 2022; Wang et al., 2022; Bahng et al., 2022; Li et al., 2024a; Yang et al., 2023) learns tokens in the visual input space while keeping the backbone frozen. (3) *Multi-modal prompt learning* (Khattak et al., 2023a; Lee et al., 2023; Li et al., 2023; Roy & Etemad, 2024; Li et al., 2024b; 2025a; Khattak et al., 2023b; Hao et al., 2025; Zheng et al., 2025; Yang et al., 2025; Yao et al., 2025; Zhang et al., 2025; Liang et al., 2026; Sun et al., 2026; Du et al., 2024a) injects prompts into both branches to enhance cross-modal interaction. A persistent challenge, particularly in multimodal designs, is that prompts encode environment-specific shortcuts, undermining robustness under distribution shift. Recent causality-inspired work attempts to address this via invariance (Gao & Dong, 2025).

**Causality and Robust Generalization.** Extensive work attributes failures under distribution shift to spurious correlations that vary across environments, advocating for predictors that rely on stable (causal) signals. Representative directions include invariant learning (*e.g.*, IRM) (Arjovsky et al., 2019), worst-group optimization (*e.g.*, GroupDRO) (Sagawa et al., 2020), and risk extrapolation (REx) (Krueger et al., 2021), all aiming to reduce dependence on environmental shortcuts. Large-scale testbeds such as DomainBed further highlight that shortcut learning and model selection remain central obstacles for reliable domain generalization (Gulrajani & Lopez-Paz, 2021). Robustness concerns are also prominent in specialized visual domains such as remote sensing, where deployable adversarial patches can substantially degrade classification models (Huang et al., 2024). These causality- and invariance-inspired perspectives suggest a key implication for prompt learning: if prompts can freely transmit environment-dependent information, they may become a shortcut channel that undermines robustness. Motivated by the above, recent work has started to incorporate causal reasoning into prompt learning for VLMs, with the goal of preventing prompts from capturing spurious correlations and improving generalization under shift (Gao & Dong, 2025). However, existing causal prompting efforts are typically built on *deterministic* prompts and do not explicitly regulate the information capacity of the prompt pathway, leaving room for shortcut leakage even when invariance is

encouraged. Do-Prompt connects causal invariance with an explicit capacity control mechanism. We introduce a *variational prompt bottleneck* to limit how much input information can be transmitted through prompts via an information-constrained latent channel (Du et al., 2020), and we couple it with *prompt-level interventions* that operationalize *do*-style perturbations directly in prompt space. This combination provides both a principled information constraint and an intervention-driven training signal, enabling robust few-shot and zero-shot classification with multi-modal prompts.

## 3. Preliminaries

### 3.1. Revisiting CLIP

Our method builds on CLIP (Radford et al., 2021), a pre-trained vision–language model consisting of a vision encoder and a text encoder that map images and texts into a shared embedding space. Following common prompt-learning setups (Zhou et al., 2022a;b; Khattak et al., 2023a; Guo & Gu, 2025), we use a ViT-based CLIP backbone and adapt it via prompt tuning while keeping the backbone parameters frozen.

**Image Encoding.** Given an image $I \in \mathbb{R}^{H \times W \times 3}$, the vision encoder splits $I$ into $B$ non-overlapping patches and projects them into patch embeddings $e_0 \in \mathbb{R}^{B \times d_v}$. A learnable class token $c_0 \in \mathbb{R}^{1 \times d_v}$ is prepended to the patch embeddings. The vision transformer has $K$ transformer blocks $\{V_i(\cdot)\}_{i=0}^{K-1}$. The $i$-th block updates the class token and patch embedding via $[c_{i+1}, e_{i+1}] = V_i([c_i, e_i])$ with $i = 0, 1, \ldots, K - 1$. The final image feature $f_x$ is obtained by projecting the last-layer class token into the shared vision–language space via $f_x = F_{\text{img}}(c_K) \in \mathbb{R}^{d_{vl}}$.

**Text Encoding.** For a tokenized text sequence of length $N$, the text encoder embeds tokens into $w_0 = [w_0^1, \ldots, w_0^N] \in \mathbb{R}^{N \times d_l}$ and processes them with $K$ transformer blocks $\{L_i(\cdot)\}_{i=0}^{K-1}$ via $w_{i+1} = L_i(w_i)$ with $i = 0, 1, \ldots, K - 1$. We use the final embedding of the last token (*e.g.*, [EOS]) and project it to the shared space: $t = F_{\text{txt}}(w_K^N) \in \mathbb{R}^{d_{vl}}$.

**Zero-shot Classification.** CLIP performs zero-shot classification by pairing each class label with a handcrafted prompt template (*e.g.*, "a photo of a {class}") and encoding it to obtain class text features $\{t_y\}_{y=1}^C$, where $C$ denotes the total number of classes. Given an image feature $f_x$, the predicted label is obtained via a cosine similarity measure between $f_x$ and each class text feature $t_y$, scaled with a temperature $\tau$:

$$p(y \mid I) = \frac{\exp(\text{sim}(f_x, t_y)/\tau)}{\sum_{i=1}^C \exp(\text{sim}(f_x, t_i)/\tau)}, \hat{y} = \arg\max_y p(y \mid I).$$
(1)

### 3.2. Multi-Modal Prompt Learning

Prompt tuning adapts CLIP by inserting a small set of learnable tokens into the transformer input, while keeping the backbone frozen. Multi-modal prompt learning (Khattak et al., 2023a; Yang et al., 2024; Guo & Gu, 2025) extends text-only prompting (Zhou et al., 2022a;b) by introducing prompts in *both* text and vision branches to better coordinate cross-modal representations for downstream classification.

Following prior work, we select a contiguous range of transformer blocks for tuning. Starting from the $J$-th block, we tune $H$ consecutive blocks, *i.e.*, $i \in \{J, \ldots, J + H - 1\}$, while leaving the remaining blocks unchanged. MaPLe (Khattak et al., 2023a) typically places prompts in shallow layers, whereas MMRL (Guo & Gu, 2025) injects prompts into deeper layers.

**Deep Language Prompting.** For each tuned layer $i \in \{J, \ldots, J + H - 1\}$, we introduce $M$ learnable prompt tokens $z_i^t \in \mathbb{R}^{M \times d_l}$ and prepend them to the token sequence $\bar{w}_i = [z_i^t, w_i] \in \mathbb{R}^{(M+N) \times d_l}$. The text transformer processes $\bar{w}_i$ and outputs $\bar{w}_{i+1} = L_i(\bar{w}_i)$. We then keep only the updated "original" token part as the input to the next layer:

$$w_{i+1} = \text{Tail}_N(\bar{w}_{i+1}), \quad i = J, \ldots, J + H - 1, \quad (2)$$

where $\text{Tail}_N(\cdot)$ extracts the last $N$ tokens. For layers outside the tuned range, the computation remains the standard CLIP update $w_{i+1} = L_i(w_i)$. The final text representation is obtained via $t = F_{\text{txt}}(w_K^N)$.

**Deep Vision Prompting.** Analogously, in the vision branch, we insert $M$ prompt tokens $z_i^v \in \mathbb{R}^{M \times d_v}$ into tuned layers. At layer $i$, the input sequence becomes $\bar{e}_i = [c_i, e_i, z_i^v]$, and the vision transformer produces

$$[c_{i+1}, e_{i+1}, \_] = V_i(\bar{e}_i), \quad i = J, \ldots, J + H - 1, \quad (3)$$

where "_" denotes outputs corresponding to prompt tokens which are not required for the subsequent projection. For layers outside the tuned range, we keep the original update $[c_{i+1}, e_{i+1}] = V_i([c_i, e_i])$. The final image feature is obtained via $f_x = F_{\text{img}}(c_K)$, and predictions follow Eq. (1).

## 4. Method

### 4.1. Overview

The goal of this work is to achieve robust, parameter-efficient adaptation of a frozen CLIP model for $C$-way classification. A core challenge in this setup arises not from insufficient model capacity, but from inductive bias in prompt learning. Because prompts form an additional pathway into the model, optimizing them on limited training environments can make them rely on environment-specific spurious

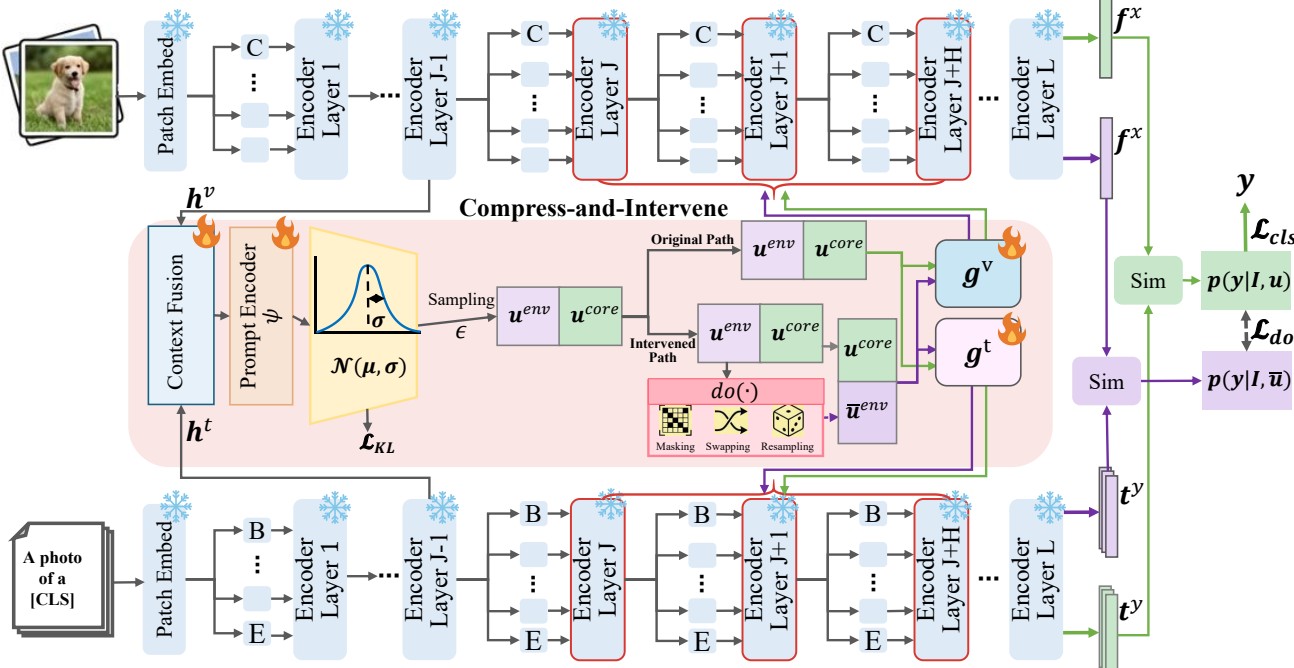

*Figure 2.* **Do-Prompt overview.** A shallow context $\boldsymbol{h}$ from frozen layers $(0, \ldots, J-1)$ parameterizes a latent prompt distribution $q_\psi(\boldsymbol{u} \mid \boldsymbol{h})$, regularized by the KL bottleneck $\mathcal{L}_{\mathrm{KL}}$. We split $\boldsymbol{u}$ into $\boldsymbol{u}^{\mathrm{core}}$ and $\boldsymbol{u}^{\mathrm{env}}$, perturb only $\boldsymbol{u}^{\mathrm{env}}$ (mask/swap/resample), and map both paths to modality-specific prompts via $g^v$ and $g^t$ for injection into tuned layers. Training combines $\mathcal{L}_{\mathrm{cls}}$ and the consistency loss $\mathcal{L}_{\mathrm{do}}$ to promote invariant semantics. Here, 'C' denotes the class token, 'B' the BOT token, and 'E' the EOT token.

correlations that fail to generalize to out-of-distribution scenarios. This issue is especially pronounced for multi-modal and input-conditioned prompting.

Do-Prompt is designed to make this prompt pathway both *information constrained* and *distributionally stable*. First, we formulate prompts as stochastic latent variables and constrain their information capacity with a variational prompt bottleneck, explicitly limiting their ability to encode environment-specific spurious cues. Second, we apply prompt-level causal interventions that perturb the environment-related part of the prompt embedding and enforce prediction consistency across these do-style perturbations. Together, these two complementary designs effectively mitigate the model's reliance on spurious shortcuts while preserving the efficiency and plug-and-play flexibility inherent to prompt tuning.

### 4.2. Variational Prompt Bottleneck

Deterministic prompts tend to absorb any signal that minimizes the training loss, including environment-specific spurious correlations, hindering cross-domain transferability. As shown in Fig. 2, we address this limitation by modeling prompts as stochastic latent variables and constraining their information capacity through a variational bottleneck. The Variational Prompt Bottleneck explicitly regulates the information flow within prompts, constraining their propensity

to encode environment-specific cues.

**Latent prompt and modality-specific projections.** For each tuned layer $i \in \{J, \ldots, J+H-1\}$, we introduce a shared latent prompt $\boldsymbol{u}_i \in \mathbb{R}^{M \times d_s}$, and obtain modality-specific prompt tokens via lightweight projectors:

$$\boldsymbol{z}_i^t = g_i^t(\boldsymbol{u}_i) \in \mathbb{R}^{M \times d_l}, \qquad \boldsymbol{z}_i^v = g_i^v(\boldsymbol{u}_i) \in \mathbb{R}^{M \times d_v}, \quad (4)$$

where $g_i^t$ and $g_i^v$ are parameter-efficient MLPs that act as projectors, mapping representations from the compact latent manifold of dimension $d_s$ to the high-dimensional text ($d_l$) and vision ($d_v$) embedding spaces, respectively. By enforcing the VIB constraint on the shared $\boldsymbol{u}_i$, we implicitly filter out non-essential information from both modalities.

**Shallow context for sample-conditioned inference.** To infer sample-specific prompts without circulant dependencies, we condition the posterior on shallow representations computed *before* the tuned range. Specifically, we extract a visual context from the vision encoder and a class-aggregated text context from the text encoder:

$$\boldsymbol{h} \triangleq \mathrm{Fuse}(\boldsymbol{h}^v, \boldsymbol{h}^t), \quad (5)$$

where $\boldsymbol{h}^v \triangleq \boldsymbol{c}_J$ with $\boldsymbol{c}_J$ being the class token at layer $J$, $\boldsymbol{h}^t \triangleq \mathrm{Pool}\left(\{\boldsymbol{w}_J^N(y)\}_{y=1}^C\right)$, $\boldsymbol{w}_J^N(y)$ is the [EOS] embedding at layer $J$ for class $y$, and $\mathrm{Pool}(\cdot)$ is a simple pooling

operator (*e.g.*, mean over classes), and $\mathrm{Fuse}(\cdot)$ can be concatenation followed by a linear layer.

**Variational posterior and sampling.** We model the posterior of each layer prompt as a diagonal Gaussian:

$$q_\phi(\boldsymbol{u}_i \mid \boldsymbol{h}) = \mathcal{N}\Big(\boldsymbol{\mu}_i(\boldsymbol{h}), \, \mathrm{diag}(\boldsymbol{\sigma}_i^2(\boldsymbol{h}))\Big), \qquad (6)$$

where $[\boldsymbol{\mu}_i(\boldsymbol{h}), \log \boldsymbol{\sigma}_i(\boldsymbol{h})] = \psi_i(\boldsymbol{h})$, and $\psi_i(\cdot)$ is a lightweight inference head and $\phi$ denotes all variational parameters. We sample $\boldsymbol{u}_i$ using reparameterization trick:

$$\boldsymbol{u}_i = \boldsymbol{\mu}_i(\boldsymbol{h}) + \boldsymbol{\sigma}_i(\boldsymbol{h}) \odot \boldsymbol{\epsilon}, \qquad \boldsymbol{\epsilon} \sim \mathcal{N}(\boldsymbol{0}, \boldsymbol{I}). \quad (7)$$

**Bottleneck regularization and ELBO view.** We place a standard normal prior $p(\boldsymbol{u}_i) = \mathcal{N}(\boldsymbol{0}, \boldsymbol{I})$ and regularize the posterior with a Kullback–Leibler (KL) divergence loss:

$$\mathcal{L}_{\mathrm{KL}} = \sum_{i=J}^{J+H-1} \mathrm{KL}(q_\phi(\boldsymbol{u}_i \mid \boldsymbol{h}) \,\|\, p(\boldsymbol{u}_i)). \qquad (8)$$

Combined with the classification negative log-likelihood, this corresponds to maximizing a variational lower bound on $\log p(y \mid I)$ with $\{\boldsymbol{u}_i\}$ as latent prompts, while explicitly limiting the information carried by the prompt channel.

### 4.3. Prompt-level Causal Interventions

The variational prompt bottleneck constrains how much information can pass through prompts, but it does not specify *which* information should be ignored. We therefore add a second ingredient that directly targets environment-dependent prompt content through *do*-style perturbations, and train the model to be stable under such perturbations.

**Core–environment split in latent prompt space.** For each tuned layer $i \in \{J, \ldots, J + H - 1\}$, we split the sampled latent prompt $\boldsymbol{u}_i$ (Sec. 4.2) into two token groups:

$$\boldsymbol{u}_i = [\boldsymbol{u}_i^{\mathrm{core}}; \boldsymbol{u}_i^{\mathrm{env}}], \qquad \text{with } M_c + M_e = M, \quad (9)$$

where $\boldsymbol{u}_i^{\mathrm{core}} \in \mathbb{R}^{M_c \times d_s}$ is intended to capture task-stable information, while $\boldsymbol{u}_i^{\mathrm{env}} \in \mathbb{R}^{M_e \times d_s}$ provides the flexibility to absorb environment/style variations. Importantly, both parts still go through the same variational bottleneck, so their capacity is controlled by the KL regularization in Eq. (8).

**Do-style interventions on $\boldsymbol{u}^{\mathrm{env}}$.** Given a sampled prompt $\boldsymbol{u}_i$, we construct an intervened counterpart $\bar{\boldsymbol{u}}_i$ by keeping $\boldsymbol{u}_i^{\mathrm{core}}$ fixed and perturbing only $\boldsymbol{u}_i^{\mathrm{env}}$ to form $\bar{\boldsymbol{u}}_i^{\mathrm{env}}$. We consider three operators:

$$\textbf{Mask:} \quad \bar{\boldsymbol{u}}_i^{\mathrm{env}} = \boldsymbol{0}, \qquad (10)$$

$$\textbf{Swap:} \quad \bar{\boldsymbol{u}}_{i,b}^{\mathrm{env}} = \boldsymbol{u}_{i,\pi(b)}^{\mathrm{env}}, \qquad (11)$$

$$\textbf{Resample:} \quad \bar{\boldsymbol{u}}_i^{\mathrm{env}} \sim \mathcal{N}(\boldsymbol{0}, \boldsymbol{I}), \qquad (12)$$

where $b$ indexes samples in a mini-batch and $\pi$ is a random permutation.

We obtain intervened modality-specific prompts (*e.g.*, $\tilde{\boldsymbol{z}}_i^{\mathrm{t}}, \tilde{\boldsymbol{z}}_i^{\mathrm{v}}$) through the same projectors as in Eq. (4). These perturbations approximate $do(\mathrm{env})$ operations $\bar{\boldsymbol{u}}_i = [\boldsymbol{u}_i^{\mathrm{core}}; \bar{\boldsymbol{u}}_i^{\mathrm{env}}]$ which disrupt the prompt portion most likely to carry environment-specific shortcuts, while preserving the task-stable part.

**Intervention consistency.** Let $p_\theta(y \mid I; \{\boldsymbol{u}_i\})$ denote the predictive distribution produced by CLIP with injected prompts $\{\boldsymbol{u}_i\}$, where $\theta$ contains only prompt-related modules with the CLIP backbone being frozen. We encourage invariance by enforcing that predictions remain consistent after intervening on $\boldsymbol{u}^{\mathrm{env}}$:

$$\mathcal{L}_{\mathrm{do}} = \mathbb{E}_k \Big[ \mathrm{KL}\Big( p_\theta(y \mid I; \{\boldsymbol{u}_i\}) \,\|\, p_\theta(y \mid I; \{\bar{\boldsymbol{u}}_i^{(k)}\}) \Big) \Big], \tag{13}$$

where $k$ indexes the intervention operator and $\bar{\boldsymbol{u}}_i^{(k)}$ is the intervened prompt. This term directly penalizes reliance on information that is not preserved under prompt-level interventions.

**Choice and interpretation of the core–environment split.** In practice, we use a fixed token partition for the latent prompt, where $u_i^{\mathrm{core}}$ is kept unchanged and only $u_i^{\mathrm{env}}$ is perturbed in the intervened path. Unless otherwise specified, we adopt a $M_c : M_e = 3 : 2$ split when $M = 5$, or the closest rounded ratio for other prompt lengths. This design intentionally allocates a larger portion to the invariant branch, so that $u_i^{\mathrm{core}}$ serves as the main carrier of stable task information, while $u_i^{\mathrm{env}}$ remains a bounded perturbable subspace for intervention.

We emphasize that this index-based partition is not intended to identify ground-truth causal variables. Instead, it defines an asymmetric regularization structure in latent prompt space: the shared KL bottleneck constrains the overall prompt capacity, while the env-only intervention consistency discourages the classifier from relying on perturbation-sensitive prompt content. Therefore, Do-Prompt should be understood as an intervention-oriented robustness regularizer, rather than a formal causal disentanglement method.

### 4.4. Training objective

Given $\{\boldsymbol{u}_i\}$ sampled from $q_\phi(\boldsymbol{u}_i \mid \boldsymbol{h})$ as in Eq. (7), we compute class probabilities with Eq. (1) and optimize the standard cross-entropy:

$$\mathcal{L}_{\mathrm{cls}} = -\mathbb{E}_{(I,y)} \log p_\theta(y \mid I; \{\boldsymbol{u}_i\}). \qquad (14)$$

The full objective combines classification, the KL bottleneck from Eq. (8), and the intervention consistency:

$$\mathcal{L} = \mathcal{L}_{\text{cls}} + \beta\,\mathcal{L}_{\text{KL}} + \lambda\,\mathcal{L}_{\text{do}}, \qquad (15)$$

where $\beta$ controls the bottleneck strength and $\lambda$ balances intervention consistency.

When the standard variation of the Gaussian $\boldsymbol{\sigma}_i(\boldsymbol{h}) \to \boldsymbol{0}$, the variational prompts are then degenerated to deterministic prompts, and $\lambda = 0$ means no interventions applied, Eq. (15) reduces to standard multi-modal prompt learning objectives used in MaPLe/MMRL-style pipelines (Khattak et al., 2023a; Guo & Gu, 2025).

## 5. Experiments

### 5.1. Experiments Setup

We evaluate the performance of our model under three different settings: base-to-novel generalization, cross-dataset evaluation, and domain generalization. All were conducted under a 16-shot setting, where each category has only 16 training examples.

**Base-to-Novel Generalization.** In this setting, dataset classes are split into base and novel classes. The model is trained on base classes and tested on both base and novel classes, enabling an assessment of its transfer learning performance on base classes and its ability to preserve the inherent generalization and zero-shot capabilities of pre-trained VLMs for novel classes. Experiments are conducted on 11 diverse classification datasets: ImageNet (Deng et al., 2009), Caltech101 (Fei-Fei et al., 2004), OxfordPets (Parkhi et al., 2012), StanfordCars (Krause et al., 2013), Flowers102 (Nilsback & Zisserman, 2008), Food101 (Bossard et al., 2014), FGVCAircraft (Maji et al., 2013), SUN397 (Xiao et al., 2010), UCF101 (Soomro et al., 2012), DTD (Cimpoi et al., 2014), and EuroSAT (Helber et al., 2019).

**Cross-Dataset Evaluation.** This evaluation examines the model's ability to generalize to unseen datasets. Following CoCoOp (Zhou et al., 2022b), the model is trained on all 1000 ImageNet classes in a few-shot setting and directly tested, without further fine-tuning, on the remaining 10 datasets used for base-to-novel generalization, allowing us to assess cross-dataset transferability.

**Domain Generalization.** To evaluate the model's robustness to domain shifts and its generalization to out-of-distribution data, we train it on ImageNet and test it on four domain-variant datasets: ImageNetV2 (Recht et al., 2019), ImageNet-Sketch (Wang et al., 2019), ImageNet-A (Hendrycks et al., 2021b), and ImageNet-R (Hendrycks et al., 2021a), which introduce diverse domain variations.

**Implementation Details.** We follow standard multi-modal prompt learning protocols (Khattak et al., 2023a; Guo & Gu, 2025) and use a 16-shot setting unless stated otherwise. We adopt CLIP (ViT-B/16) as the frozen backbone and inject prompts into a contiguous range of transformer blocks, from layer $J$ to $J + H - 1$, in both the vision and text encoders. We report results under two configurations: MMRL-style ($J{=}5$, $H{=}7$, $M{=}5$) and MaPLe-style ($J{=}0$, $H{=}9$, $M{=}2$). Do-Prompt follows the same prompt placement as the corresponding baseline and adds only the variational bottleneck and prompt-level interventions. All experiments are conducted on a single NVIDIA V100 GPU. Additional implementation details are provided in the Appendix A.

### 5.2. Base to Novel Generalization

Table 1 evaluates the generalization of the proposed method by training on base classes and testing on novel classes across 11 datasets. The results consistently demonstrate that integrating **Do-Prompt** into multi-modal learners, such as MaPLe and MMRL, yields a superior trade-off between base-class proficiency and novel-class adaptability. Specifically, Do-Prompt provides a significant performance lift to the aggregate Harmonic Mean (HM), boosting MaPLe by **1.75%** and MMRL by **1.97%** on average. These improvements are not incremental; they reflect a fundamental enhancement in the model's ability to extract stable, transferable features rather than overfitting to base-set biases. The advantages of the proposed approach are most pronounced in specialized domains where the distribution shift between base and novel categories is challenging. For instance, on the **EuroSAT** and **DTD** datasets, which are prone to dataset-specific shortcut cues, Do-Prompt achieves remarkable HM gains (*e.g.*, a **+5.78%** improvement for MMRL on EuroSAT). These results validate our core hypothesis: by employing a variational bottleneck to filter non-causal artifacts and utilizing prompt-level interventions, Do-Prompt steers the model toward robust evidence valid in unseen scenarios.

### 5.3. Cross-Dataset Transfer

Table 2 demonstrates that Do-Prompt significantly enhances out-of-distribution transferability, yielding average gains of **+1.56%** for MaPLe and **+1.77%** for MMRL. These improvements are consistent across diverse domains, with a standout increase reaching **+9.20%** on the specialized **EuroSAT** dataset. Such results indicate that the proposed variational interventions successfully suppress source-specific shortcuts, forcing the model to rely on stable, task-relevant evidence that generalizes far beyond the training distribution.

### 5.4. Domain Generalization

Table 3 demonstrates that Do-Prompt significantly enhances model robustness against distribution shifts, providing consistent gains across all ImageNet variants. Notably, the proposed method yields substantial improvements for MMRL,

*Table 1.* **Base to novel generalization results**. Overall, our Do-Prompt method improves the base performance of both MaPLe and MMRL, with a considerable increase in FGVCAircraft and EuroSAT datasets. An asterisk (*) denotes methods we re-ran using the same random seeds and hardware. Delta values (Δ) show improvements from adding Do-Prompt (**red** for positive, **blue** for negative).

| Method | Average | | | ImageNet | | | Caltech101 | | | OxfordPets | | |
|---|---|---|---|---|---|---|---|---|---|---|---|---|
| | Base | Novel | HM | Base | Novel | HM | Base | Novel | HM | Base | Novel | HM |
| CLIP (Radford et al., 2021) | 69.34 | 74.22 | 71.70 | 72.43 | 68.14 | 70.22 | 96.84 | 94.00 | 95.40 | 91.17 | 97.26 | 94.12 |
| CoOp (Zhou et al., 2022a) | 82.69 | 63.22 | 70.83 | 76.47 | 67.88 | 71.92 | 98.00 | 89.81 | 93.73 | 93.67 | 95.29 | 94.47 |
| CoCoOp (Zhou et al., 2022b) | 80.47 | 71.69 | 75.83 | 75.98 | 70.43 | 73.10 | 97.96 | 93.81 | 95.84 | 95.20 | 97.69 | 96.43 |
| APP (Cho et al., 2024) | 83.0 | 65.8 | 72.61 | 69.9 | 63.2 | 66.4 | 95.2 | 91.0 | 93.0 | 96.8 | 88.3 | 92.4 |
| PromptSRC* (Khattak et al., 2023b) | 84.93 | 74.49 | 78.61 | 76.77 | 67.8 | 72.01 | 98.07 | 94.03 | 96.01 | 95.27 | 97.23 | 96.24 |
| MaPLe* (Khattak et al., 2023a) | 82.03 | 75.03 | 78.37 | 74.96 | 66.97 | 70.74 | 97.83 | 94.87 | 96.33 | 95.20 | 98.13 | 96.64 |
| **MaPLe* + Do-Prompt (Ours)** | **83.82** | **76.68** | **80.12** | 76.06 | 68.80 | 72.25 | 98.35 | 94.87 | 96.58 | 96.17 | 97.77 | **96.96** |
| Improvements Δ | +1.79 | +1.65 | +1.75 | +1.10 | +1.83 | +1.51 | +0.52 | 0.0 | +0.25 | +0.97 | -0.36 | +0.32 |
| MMRL* (Guo & Gu, 2025) | 85.54 | 76.52 | 80.59 | 77.55 | 67.43 | 72.14 | 98.93 | 94.60 | 96.72 | 95.27 | 97.23 | 96.24 |
| **MMRL* + Do-Prompt (Ours)** | **86.38** | **79.16** | **82.56** | 77.90 | 68.83 | 73.09 | 99.10 | 94.27 | 96.62 | 95.93 | 97.57 | 96.74 |
| Improvements Δ | +0.84 | +2.64 | +1.97 | +0.35 | +1.40 | +0.95 | +0.17 | -0.33 | -0.10 | +0.66 | +0.34 | +0.50 |

| Method | StanfordCars | | | Flowers102 | | | Food101 | | | FGVCAircraft | | |
|---|---|---|---|---|---|---|---|---|---|---|---|---|
| | Base | Novel | HM | Base | Novel | HM | Base | Novel | HM | Base | Novel | HM |
| CLIP (Radford et al., 2021) | 63.37 | 74.89 | 68.65 | 72.08 | 77.80 | 74.83 | 90.10 | 91.22 | 90.66 | 27.19 | 36.29 | 31.09 |
| CoOp (Zhou et al., 2022a) | 78.12 | 60.40 | 68.13 | 97.60 | 59.67 | 74.06 | 88.33 | 82.26 | 85.19 | 40.44 | 22.30 | 28.75 |
| CoCoOp (Zhou et al., 2022b) | 70.49 | 73.59 | 72.01 | 94.87 | 71.75 | 81.71 | 90.70 | 91.29 | 90.99 | 33.41 | 23.71 | 27.74 |
| APP (Cho et al., 2024) | 85.9 | 69.5 | 76.8 | 96.8 | 61.0 | 74.8 | 84.6 | 86.1 | 85.4 | 44.9 | 26.0 | 33.0 |
| PromptSRC* (Khattak et al., 2023b) | 77.93 | 75.5 | 76.7 | 97.83 | 77.13 | 86.26 | 90.60 | 91.53 | 91.06 | 41.43 | 23.67 | 30.13 |
| MaPLe* (Khattak et al., 2023a) | 72.45 | 74.90 | 73.65 | 96.33 | 73.33 | 83.27 | 90.80 | 92.10 | 91.45 | 36.60 | 34.90 | 35.73 |
| **MaPLe* + Do-Prompt (Ours)** | 74.73 | 74.57 | 74.65 | 97.43 | 74.37 | 84.35 | **90.83** | **92.13** | 91.48 | **39.80** | **36.63** | **38.15** |
| Improvements Δ | +2.28 | -0.33 | +1.0 | +1.10 | +1.04 | +1.08 | +0.03 | +0.03 | +0.03 | +3.20 | +1.73 | +2.42 |
| MMRL (Guo & Gu, 2025) | **81.23** | 75.00 | 77.99 | 98.70 | 76.83 | 86.40 | 90.60 | 91.53 | 91.06 | **45.70** | 37.1 | 40.95 |
| **MMRL* + Do-Prompt (Ours)** | 81.20 | **75.37** | **78.17** | **98.80** | **77.23** | **86.70** | 90.73 | 91.60 | 91.16 | 47.13 | 40.57 | 43.60 |
| Improvements Δ | -0.03 | +0.37 | +0.18 | +0.10 | +0.40 | +0.30 | +0.13 | +0.07 | +0.10 | +1.43 | +3.47 | +2.65 |

| Method | SUN397 | | | DTD | | | EuroSAT | | | UCF101 | | |
|---|---|---|---|---|---|---|---|---|---|---|---|---|
| | Base | Novel | HM | Base | Novel | HM | Base | Novel | HM | Base | Novel | HM |
| CLIP (Radford et al., 2021) | 69.36 | 75.35 | 72.23 | 53.24 | 59.90 | 56.37 | 56.48 | 64.05 | 60.03 | 70.53 | 77.50 | 73.85 |
| CoOp (Zhou et al., 2022a) | 80.60 | 65.89 | 72.51 | 79.44 | 41.18 | 54.24 | 92.19 | 54.74 | 68.69 | 84.69 | 56.05 | 67.46 |
| CoCoOp (Zhou et al., 2022b) | 79.74 | 76.86 | 78.27 | 77.01 | 56.00 | 64.85 | 87.49 | 60.04 | 71.21 | 82.33 | 73.45 | 77.64 |
| APP (Cho et al., 2024) | 80.6 | 73.3 | 76.8 | 78.4 | 48.9 | 60.2 | 93.6 | 47.6 | 63.1 | 86.2 | 69.2 | 76.8 |
| PromptSRC* (Khattak et al., 2023b) | 92.93 | 74.17 | 80.53 | 83.33 | 61.3 | 70.64 | 92.93 | 74.17 | 82.5 | 87.10 | 78.53 | 82.6 |
| MaPLe* (Khattak et al., 2023a) | 81.07 | 77.90 | 79.45 | 81.30 | 57.50 | 67.36 | 92.47 | 77.03 | 84.05 | 83.97 | 77.70 | 80.71 |
| **MaPLe* + Do-Prompt (Ours)** | 81.83 | **79.87** | 80.84 | 83.50 | 63.60 | 72.21 | 96.30 | 81.20 | 88.11 | 84.97 | 79.73 | 82.27 |
| Improvements Δ | +0.76 | +1.97 | +1.39 | +2.2 | +6.10 | +4.85 | +3.83 | +4.17 | +4.06 | +1.0 | +2.03 | +1.56 |
| MMRL (Guo & Gu, 2025) | 83.23 | 79.27 | 81.20 | 85.63 | **65.13** | 73.99 | 95.80 | 77.20 | 85.50 | 88.30 | 79.70 | 83.78 |
| **MMRL* + Do-Prompt (Ours)** | **83.20** | 79.37 | **81.24** | **86.80** | 67.50 | **75.94** | **97.73** | **85.63** | **91.28** | **88.33** | **80.40** | **84.18** |
| Improvements Δ | -0.03 | +0.10 | +0.04 | +1.17 | +2.37 | +1.95 | +1.93 | +8.43 | +5.78 | +0.03 | +0.70 | +0.40 |

*Table 2.* **Comparison of the proposed Do-Prompt method with previous state-of-the-art multi-modal prompt learning methods on cross-dataset evaluation across 10 datasets.**

| | Source | Target | | | | | | | | | | |
|---|---|---|---|---|---|---|---|---|---|---|---|---|
| | ImageNet | Caltech101 | OxfordPets | StanfordCars | Flowers102 | Food101 | FGVCAircraft | SUN397 | DTD | EuroSAT | UCF101 | Average |
| PromptSRC* (Khattak et al., 2023b) | 68.96 | 93.47 | 90.33 | 65.87 | 70.40 | 83.66 | 24.13 | 67.17 | 46.40 | 45.97 | 67.67 | 65.50 |
| MaPLe* (Khattak et al., 2023a) | 67.96 | 93.17 | 90.20 | 65.97 | 71.07 | 86.33 | 23.23 | 67.23 | 47.20 | 45.70 | 66.27 | 65.63 |
| **MaPLe* + Do-Prompt (Ours)** | **68.66** | **93.80** | **90.73** | **66.30** | **74.57** | **86.40** | **25.37** | **67.70** | **48.17** | **54.90** | **70.87** | **67.19** |
| Improvements Δ | +0.70 | +0.63 | +0.53 | +0.33 | +3.50 | +0.07 | +2.14 | +0.47 | +0.97 | +9.20 | +4.60 | +1.56 |
| MMRL* (Guo & Gu, 2025) | 70.13 | 94.30 | 91.00 | 66.20 | 71.53 | 86.20 | 26.07 | 67.50 | 46.93 | 49.83 | 69.13 | 66.87 |
| **MMRL* + Do-Prompt (Ours)** | **71.00** | **94.60** | **91.97** | **66.63** | **75.57** | **86.43** | **28.33** | **67.90** | **48.27** | **58.97** | **72.50** | **68.64** |
| Improvements Δ | +0.87 | +0.30 | +0.97 | +0.43 | +4.04 | +0.23 | +2.26 | +0.40 | +1.34 | +9.14 | +3.37 | +1.77 |

with performance boosts reaching up to **+1.90%** on challenging shifts like ImageNet-Sketch and ImageNet-A. These results validate that by filtering non-causal artifacts through the proposed variational bottleneck and enforcing intervention consistency, Do-Prompt learns inherently more stable prompts that resist degradation under unseen domain shifts.

## 5.5. Ablations

We conduct ablation studies to analyze (i) the contributions of each component, and (ii) robustness across prompt backbones (MaPLe vs. MMRL) and evaluation protocols.

**Key components.** Table 4 demonstrates the effectiveness of both Do-Prompt components. Employing either the variational prompt bottleneck or the do-intervention individually

*Table 3.* **Comparison of robustness on out-of-distribution datasets.** Delta values ($\Delta$) show improvements from Do-Prompt.

| Method | Source | Target | | | |
|---|---|---|---|---|---|
| | ImageNet | ImageNetV2 | ImageNet-S | ImageNet-A | ImageNet-R |
| PromptSRC* (Khattak et al., 2023b) | 68.96 | 62.50 | 48.60 | 49.63 | 75.77 |
| MaPLe* (Khattak et al., 2023a) | 67.96 | 61.57 | 47.70 | 48.80 | 75.33 |
| **MaPLe* + Do-Prompt (Ours)** | **68.66** | **62.80** | **49.20** | **50.30** | **76.53** |
| Improvements $\Delta$ | +0.70 | +1.23 | +1.50 | +1.50 | +1.20 |
| MMRL* (Guo & Gu, 2025) | 70.13 | 62.20 | 47.80 | 48.90 | 75.03 |
| **MMRL* + Do-Prompt (Ours)** | **71.00** | **63.80** | **49.70** | **50.80** | **76.83** |
| Improvements $\Delta$ | +0.87 | +1.60 | +1.90 | +1.90 | +1.80 |

*Table 4.* **Ablation on key components.** We report the average Base, Novel, and HM accuracy across 11 datasets.

| Variant | VIB | do-int. | Base | Novel | HM |
|---|---|---|---|---|---|
| MaPLe* baseline | | | 82.03 | 75.03 | 78.37 |
| + VIB only | ✓ | | 82.71 | 75.94 | 78.93 |
| + do-intervention only | | ✓ | 82.84 | 76.01 | 79.02 |
| **MaPLe* + Do-Prompt** | ✓ | ✓ | **83.82** | **76.68** | **80.12** |
| MMRL* baseline | | | 85.54 | 76.52 | 80.59 |
| + VIB only | ✓ | | 85.61 | 76.93 | 81.32 |
| + do-intervention only | | ✓ | 85.67 | 76.99 | 81.43 |
| **MMRL* + Do-Prompt** | ✓ | ✓ | **86.38** | **79.16** | **82.56** |

*Table 5.* **Ablation on intervention operators.** "All" corresponds to the default Do-Prompt (mask+swap+resample).

| Intervention | Base | Novel | Avg HM |
|---|---|---|---|
| None (MaPLe) | 82.03 | 75.03 | 78.37 |
| Mask only | 83.01 | 75.97 | 79.01 |
| Swap only | 82.78 | 75.75 | 78.94 |
| Resample only | 83.06 | 75.82 | 79.15 |
| **All (Do-Prompt)** | **83.82** | **76.68** | **80.12** |

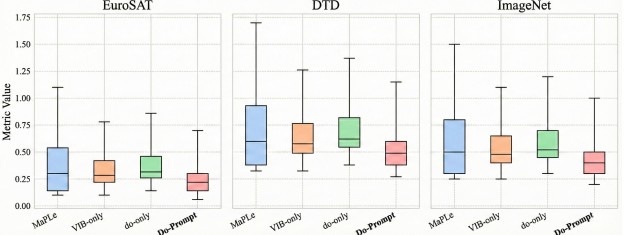

*Figure 3.* **Prediction invariance under prompt interventions.** Distribution of $\mathrm{KL}(p(y \mid I, \boldsymbol{u}) \| p(y \mid I, \bar{\boldsymbol{u}}))$ on EuroSAT, DTD, and ImageNet. Lower values indicate more stable predictions; Do-Prompt yields the smallest shift.

improves the base-to-novel balance, while combining them consistently achieves the best results on both MaPLe and MMRL. Overall, the two terms are complementary: the bottleneck explicitly constrains prompt channel capacity, while the intervention objective actively discourages reliance on environment-specific prompt content.

**Intervention operators.** Table 5 compares different intervention operators. Each operator improves over the baseline, and the full combination performs best, indicating that the interventions provide complementary perturbations in prompt space. Therefore, we use the combined setting as the default configuration in all experiments.

### 5.6. Analysis

**Invariance under prompt-level interventions.** Fig. 3 visualizes the change in the predictive distribution after intervening on the environment-related prompt component, measured by $\mathrm{KL}(p(y \mid I, \boldsymbol{u}) \| p(y \mid I, \bar{\boldsymbol{u}}))$. Across three datasets, Do-Prompt yields consistently smaller shifts, indicating its predictions are less sensitive to prompt perturbations and rely less on environment-specific content.

**Capacity control during training.** Fig. 4 reports the training dynamics of the bottleneck regularizer $\mathcal{L}_{\mathrm{KL}}$. The KL term remains stable throughout optimization, supporting that the prompt channel is effectively constrained rather than becoming an unconstrained shortcut pathway.

**Diagnostic comparison of prompt-branch energy.** Fig. 5 reports a diagnostic comparison between the two

latent prompt branches. We define the prompt energy as the average token $\ell_2$ norm:

$$E_{\mathrm{core}} = \frac{1}{|\mathcal{D}|HM_c} \sum_{I \in \mathcal{D}} \sum_{i=J}^{J+H-1} \sum_{m=1}^{M_c} \left\| u_{i,m}^{\mathrm{core}}(I) \right\|_2,$$

and compute $E_{\mathrm{env}}$ analogously over the $M_e$ environment tokens. The consistently larger value of $E_{\mathrm{core}}$ should not be interpreted as a standalone proof that this branch exactly captures task-stable semantics. Instead, together with the reduced prediction shift under interventions in Fig. 3 and the controlled prompt capacity in Fig. 4, it provides an auxiliary diagnostic that the two branches are used differently after training. This supports the weaker but more precise claim that $u^{\mathrm{env}}$ behaves as a bounded, perturbation-sensitive subspace useful for robustness.

**Computational Overhead.** Table 6 summarizes the computational and parameter overhead of Do-Prompt when integrated with MaPLe and MMRL. The additional training cost primarily stems from the extra intervened forward pass required to compute $\mathcal{L}_{\mathrm{do}}$, resulting in approximately $1.8\times$ and $1.9\times$ training time per iteration compared to the MaPLe and MMRL baselines, respectively. However, during inference, we utilize the deterministic posterior mean without interventions. Consequently, the inference latency remains negligible, with only a marginal increase of $1.02\times$–$1.03\times$. In terms of model size, Do-Prompt introduces only lightweight prompt modules, adding just 4.0M and 7.0M trainable parameters to MaPLe and MMRL, respectively. Overall, Do-Prompt significantly improves robustness while maintaining minimal overhead for deployment.

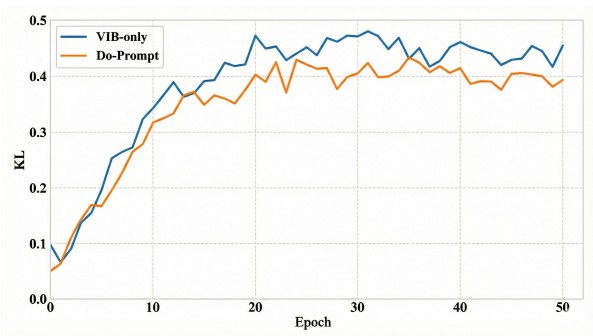

*Figure 4.* **KL bottleneck during training.** Training dynamics of the prompt bottleneck term $\mathcal{L}_{\mathrm{KL}}$ for VIB-only and Do-Prompt, showing controlled prompt capacity throughout optimization.

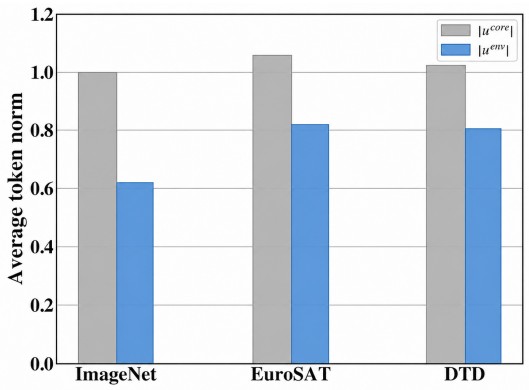

*Figure 5.* **Core vs. environment prompt energy.** Average token norm of $u^{\mathrm{core}}$ and $u^{\mathrm{env}}$ across datasets, illustrating the separation between task-stable and environment-related components.

**Sensitivity to the core–environment split.** We further study the sensitivity of Do-Prompt to the ratio between the core and environment prompt groups. As shown in Table 7, Do-Prompt is reasonably stable around the default $3 : 2$ partition. Performance drops when the environment branch becomes overly dominant, which is consistent with its intended role as a bounded perturbation subspace rather than the main carrier of task-stable information.

**Limitations** Do-Prompt is motivated by causal invariance, but it does not identify true environment variables or provide a formal guarantee of exact causal disentanglement. The core–environment split is an index-based structural partition in latent prompt space, and the do-style operators are heuristic perturbations that induce an invariance regularizer rather than literal interventions on grounded causal variables. Therefore, our empirical analyses should be interpreted as evidence of functional robustness and reduced intervention sensitivity, not as definitive semantic verification that $u^{\mathrm{core}}$ and $u^{\mathrm{env}}$ exactly correspond to task and environment factors. Stronger validation with explicit nuisance annotations, controlled style/background interventions, or semantic probing is an important direction for future work. In addition, our

*Table 6.* **Overhead of Do-Prompt.** Training overhead stems mainly from the additional intervened forward pass for $\mathcal{L}_{\mathrm{do}}$, while inference uses the posterior mean without interventions.

| Method | Train Time / Iter | Inference Time / Img | Params ($\Delta$) |
|---|---|---|---|
| MaPLe (Khattak et al., 2023a) | $1.00\times$ | $1.00\times$ | – |
| **MaPLe + Do-Prompt** | $1.8\times$ | $1.02\times$ | +4.0M |
| MMRL (Guo & Gu, 2025) | $1.00\times$ | $1.00\times$ | – |
| **MMRL + Do-Prompt** | $1.9\times$ | $1.03\times$ | +7.0M |

*Table 7.* Sensitivity to the core–environment split. We report the average harmonic mean (HM) on the base-to-novel benchmark.

| $M_c : M_e$ | **MaPLe + Do-Prompt** | **MMRL + Do-Prompt** |
|---|---|---|
| 4 : 1 | 79.31 | 81.75 |
| 3 : 2 | **80.12** | **82.56** |
| 2 : 3 | 79.27 | 81.27 |
| 1 : 1 | 79.15 | 81.09 |

experiments mainly follow the standard CLIP ViT-B/16 and few-shot prompt-learning protocol; extending Do-Prompt to larger VLM backbones, non-CLIP architectures, and broader data regimes remains future work.

## 6. Conclusion

In this paper, we presented **Do-Prompt** to improve the robustness of multi-modal prompt tuning for CLIP-style classification under distribution shift. Do-Prompt targets a practical failure mode where prompts overfit to environment-specific shortcuts. It combines a variational prompt bottleneck to constrain prompt capacity with prompt-level causal interventions that enforce prediction stability under perturbations of environment-related prompt content. The resulting compress-and-intervene framework is parameter-efficient, integrates into MaPLe/MMRL-style pipelines with minimal changes, and consistently improves base-to-novel, cross-dataset, and domain generalization performance, especially on datasets with stronger domain or texture shifts.

## Impact Statement

This paper presents work whose goal is to advance the field of Machine Learning. There are many potential societal consequences of our work, none which we feel must be specifically highlighted here.

## Acknowledgments

We sincerely thank the reviewers for their insightful comments and valuable suggestions. This work was supported by the National Natural Science Foundation of China under Grant No. 62572480 and the Provincial Natural Science Foundation of Hunan under Grant No. 2025JJ10008.

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

## A. Implementation Details

We follow the standard multi-modal prompt learning protocol and use a 16-shot setting for all datasets unless stated otherwise (Khattak et al., 2023a; Guo & Gu, 2025). All experiments are built on CLIP with a ViT-B/16 backbone (Radford et al., 2021). The CLIP image and text encoders are kept frozen throughout training and evaluation. We perform multi-layer prompt tuning by injecting prompt tokens into a contiguous range of transformer blocks, from layer $J$ to $J + H - 1$, in both the vision and text encoders.

We report results under two representative prompt configurations. For the MMRL-style setting, we set $J{=}5$ and $H{=}7$, and insert $M{=}5$ prompt tokens per tuned layer. For the MaPLe-style setting, we set $J{=}0$ and $H{=}9$ with prompt length $M{=}2$ for both modalities. Do-Prompt is implemented as a drop-in extension on top of these baselines: we first extract a shallow context $h$ from the frozen layers $(0 \ldots J{-}1)$, and then use it to parameterize a layer-wise latent prompt distribution $q_\psi(u_i \mid h)$. We split the $M$ latent prompt tokens into $M_c$ core and $M_e$ env tokens by index, with $M_c > M_e$ by default (e.g., $3 : 2$ when $M = 5$). Unless otherwise specified, we set the bottleneck weight to $\beta = 0.5$ and the intervention-consistency weight to $\lambda = 0.5$ for all datasets and settings. At inference time, we use the posterior mean $u_i = \mu_i(h)$ and do not apply sampling or interventions.

**Trainable parameters.** We only optimize the context fusion layer $(\text{Fuse}(\cdot))$, the prompt encoder $\psi$ (which outputs the variational parameters of $q_\psi$) and the modality-specific projectors $g^t$ and $g^v$ that map latent prompts to text/vision prompt tokens. All CLIP backbone parameters remain fixed. For causal interventions, we split each latent prompt into task-stable and environment-related parts and perturb only the environment part using masking, within-batch swapping, and prior resampling; the combined setting is used by default.

**Inference.** At test time, we use the posterior mean to construct prompts, i.e., we set $u_i = \mu_i(h)$ without sampling, and run a single deterministic forward pass for prediction. All experiments are conducted on a single NVIDIA V100 GPU.

## B. Additional Sensitivity Analyses

**Context simplification.** The shallow context $h$ is used only to parameterize the latent prompt posterior $q_\phi(u \mid h)$. To verify that Do-Prompt does not rely on a fragile context design, we simplify $h$ into vision-only, text-only, and class-agnostic text variants. As shown in Table 8, the full fused context performs best, while Do-Prompt remains effective under simplified variants, indicating that the main robustness gain comes from the bottleneck-plus- intervention design.

*Table 8.* Context ablation. We report average HM on the base-to-novel benchmark.

| Context for $q_\phi(u \mid h)$ | MaPLe | MMRL |
|---|---|---|
| Vision + class-aggregated text | **80.12** | **82.56** |
| Vision-only | 77.25 | 79.31 |
| Text-only | 77.93 | 79.97 |
| Class-agnostic text-only | 76.88 | 79.05 |

**Sensitivity to $\beta$ and $\lambda$.** We use $\beta = 0.5$ and $\lambda = 0.5$ as the default setting across datasets and settings. Table 9 varies one hyperparameter at a time around the default. The results remain stable within a moderate range of both $\beta$ and $\lambda$, suggesting that the gains are not caused by fragile hyperparameter tuning.

*Table 9.* Sensitivity to $\beta$ and $\lambda$. We report average HM.

| Setting | MaPLe | MMRL |
|---|---|---|
| Default ($\beta = 0.5, \lambda = 0.5$) | **80.12** | **82.56** |
| $\beta = 0.25$ | 80.12 | 82.56 |
| $\beta = 1.0$ | 80.03 | 82.05 |
| $\lambda = 0.25$ | 79.93 | 81.97 |
| $\lambda = 1.0$ | 79.98 | 82.03 |

### B.1. Additional Protocol Validation

**Cross-scale CLIP backbone.**    To verify that Do-Prompt is not tied to a single CLIP scale, we evaluate it with a larger CLIP ViT-L/14 backbone. As shown in Table 10, Do-Prompt continues to improve over the corresponding MaPLe baseline, suggesting that the compress-and-intervene design generalizes beyond the default ViT-B/16 setting.

*Table 10.* Cross-scale validation with CLIP ViT-L/14.

| Method | Base | Novel | HM |
| --- | --- | --- | --- |
| MaPLe | 83.15 | 77.25 | 80.09 |
| MaPLe + Do-Prompt | **85.72** | **80.92** | **83.25** |

**Performance beyond the 16-shot setting.**    Although our main experiments follow the standard 16-shot prompt-learning protocol, we further evaluate Do-Prompt in the more data-scarce 1-shot setting. Table 11 shows that the improvement persists, indicating that Do-Prompt is not specific to the 16-shot regime.

*Table 11.* Performance beyond the 16-shot setting. We report average HM.

| Method | 1-shot | 16-shot |
| --- | --- | --- |
| MaPLe | 71.21 | 78.37 |
| MaPLe + Do-Prompt | **75.23** | **80.12** |
| MMRL | 73.17 | 80.59 |
| MMRL + Do-Prompt | **76.94** | **82.56** |

**Posterior-mean inference.**    At test time, we use the deterministic posterior mean $u_i = \mu_i(h)$ without sampling. We also tested a small number of Monte Carlo samples at inference. Using 2 and 4 samples slightly improves the average HM from 80.12 to 80.25 and 80.43, respectively, but increases inference cost accordingly. Therefore, we keep posterior-mean inference as the default choice, since training-time stochasticity already provides most of the robustness benefit while preserving efficient deployment.

