# OpenReview forum: "Do-Prompt: Causal Interventions Meet Variational Prompt Bottlenecks"
_ICML.cc/2026/Conference — ICML 2026 regular_

### Official Review · Reviewer_HFcG · 2026-03-06

**Soundness:** 3
**Presentation:** 3
**Significance:** 3
**Originality:** 3
**Overall Recommendation:** 4
**Confidence:** 4

**Summary:**

This paper proposes Do-Prompt, a method for improving the robustness of multi-modal prompt learning for vision–language models such as CLIP. This paper indicates that prompts can act as a high-capacity channel that encodes environment-specific spurious correlations during training, which harms generalization under distribution shift. To address this, the authors introduce a compress-and-intervene framework that combines a variational prompt bottleneck and prompt-level causal interventions that perturb environment-related prompt components and enforce prediction consistency. Overall, this work studies the topic of improving prompt robustness and generalization in multi-modal prompt tuning, and demonstrates through experiments on base-to-novel generalization, cross-dataset transfer, and domain generalization benchmarks that integrating Do-Prompt into existing methods yields consistent performance improvements.

**Compliance With Llm Reviewing Policy:**

Affirmed.

**Final Justification:**

I thank the authors for the response. I am keeping my positive score.

**Key Questions For Authors:**

1. The paper splits u into u^{core} and u^{env}. However, the split is architecturally imposed rather than learned from explicit environment variables, and the interventions operate only in latent prompt space. Could the authors clarify why this should be interpreted as a causal mechanism rather than a regularization strategy?

2. The paper claims that u^{core} captures task-stable semantics and u^{env} captures environment-related variation. However, the evidence (e.g., token norm differences and KL stability) does not directly validate this decomposition. Could the authors provide additional analysis verifying that the u^{env} indeed captures environment-related information?

**Limitations:**

See weaknesses and questions.

**Strengths And Weaknesses:**

Strengths:
1. The paper addresses a limitation of prompt learning for vision–language models that learned prompts can encode environment-dependent spurious correlations that harm generalization under distribution shift. The proposed Do-Prompt framework is conceptually clean, combining a variational prompt bottleneck with prompt-level interventions in a unified “compress-and-intervene” design. The method is well-motivated and the overall architecture is clearly presented.

2. The proposed method can be easily integrated into existing multi-modal prompt learning frameworks, such as MaPLe and MMRL, without modifying the backbone model.

3. The paper evaluates the method on several challenging scenarios, including base-to-novel generalization, cross-dataset transfer, and domain generalization benchmarks. The results show consistent improvements over strong baselines across multiple datasets.

Weaknesses:
1. The paper’s causal/mechanistic claims are stronger than the evidence provided. The method is motivated by the idea that prompts encode environment-specific spurious correlations, and the paper interprets the split into u^{core} and u^{env} as separating task-stable versus environment-related information. But in the actual method, this split is imposed architecturally rather than identified from supervision or intervention on true environment variables, and the “do-style” perturbations are heuristic operations in latent prompt space (mask, swap, resample) rather than interventions on a grounded causal variable. As a result, the work demonstrates a useful regularization strategy, but not a convincing causal decomposition of prompt content.

2. The main supporting analysis is that u^{core} has a larger average token norm than u^{env}, plus reduced KL shift under interventions, but neither result really verifies that one component captures semantics and the other captures nuisance factors. There is no probing study, visualization of retrieved semantics, correlation with known nuisance attributes, or controlled experiment showing that  u^{env} tracks style/background while u^{core} preserves class information.

---

> ### Author Rebuttal · Authors · 2026-03-31
>
> *We thank the reviewer for the thoughtful and positive feedback.*
>
>
> *W1: Strength of the Causal Claims.*
>
> **A1**: We agree that the current evidence does not justify a claim of exact causal decomposition of prompt content, and our method does not identify true environment variables from supervision. Our intent is more modest: Do-Prompt uses a causality-inspired structural regularization in latent prompt space, where one subspace is kept invariant and the other is explicitly perturbed, to discourage reliance on unstable prompt factors. In this sense, the “do-style” operators are not meant to be literal interventions on grounded causal variables, but heuristic perturbations that induce an invariance objective in the prompt channel. The main empirical claim is therefore not causal identification, but that the combination of a variational bottleneck and env-only perturbation leads to more robust prompt learning, which is supported by the ablations and analyses in the paper. We will revise the wording throughout to make this distinction clearer and avoid overstating the method as a convincing causal decomposition.
>
>
> *W2: Lack of Direct Semantic Validation.*
>
> **A2**: In the current submission, the norm gap in Fig. 5 and the reduced prediction shift under intervention in Fig. 3 are intended only as supporting diagnostics, not as definitive evidence of semantic/nuisance disentanglement. Our claim is therefore narrower: these results indicate that the learned partition is functionally meaningful for robustness, in the sense that predictions become less sensitive to perturbations of the designated environment branch and the two subspaces exhibit different usage patterns. We agree that stronger evidence such as probing, visualization, or correlation with known nuisance attributes would be valuable, and we will position these as important follow-up analyses rather than overstate the current results as fully verifying the semantic roles of $u^{core}$ and $u^{env}$.
>
>
>
> *Q1: Interpretation of the Split and Interventions.*
>
> **A1**: The split into $u^{core}$ and $u^{env}$ is indeed imposed architecturally, and the interventions are heuristic perturbations in latent prompt space rather than interventions on explicit environment variables. Our use of the causal language is therefore meant in a narrower sense: the method introduces an asymmetric invariance constraint, where one subspace is kept fixed, and the other is perturbed, so that the model is discouraged from relying on unstable prompt factors. In other words, the practical contribution is not grounded causal recovery, but a structured regularization mechanism that is motivated by causal invariance principles and empirically improves robustness.  We also add a small split-ratio ablation. The results show that Do-Prompt is reasonably stable around the default setting, while performance drops when the environment branch becomes overly dominant, which is consistent with its intended role as a bounded perturbation subspace. We will revise the wording to make this distinction explicit and avoid overstating the method as a true causal mechanism. We will also include this table in the revised version.
>
> | core:env | MaPLe + Do-Prompt | MMRL + Do-Prompt |
> | ----------------- | ----------------: | ---------------: |
> | 4:1 | 79.31 | 81.75 |
> | **3:2 (default)** | **80.12** | **82.56** |
> | 2:3 | 79.27 | 81.27 |
> | 1:1 | 79.15 | 81.09 |
>
>
>
> *Q2: Evidence for $u^{env}$ as Environment-Related.*
>
> **A2**: Our current analyses are therefore intended as supporting diagnostics, not as definitive verification of semantic/nuisance disentanglement. Specifically, the smaller prediction shift under perturbing only $u^{env}$ (Fig. 3) indicates that Do-Prompt reduces reliance on that designated branch, while the norm gap between $u^{core}$ and $u^{env}$ (Fig. 5) suggests a non-trivial difference in how the two subspaces are used. Together with the fact that the intervention is applied exclusively to $u^{env}$ by design, these results support the weaker claim that $u^{env}$ behaves as a bounded, perturbation-sensitive subspace useful for robustness, rather than a fully verified representation of true environment variables. We agree that stronger validation, such as probing against known nuisance attributes or controlled style/background analyses, would be valuable, and we will revise the wording to avoid overstating the current evidence.

---

> > ### Author Rebuttal · Reviewer_HFcG · 2026-04-03
> >
> > I thank the authors for their detailed response. My concerns have been addressed, and I will maintain my positive score.

---

> > > ### Author Response · Authors · 2026-04-04
> > >
> > > Thank you for taking the time to review our rebuttal and for maintaining your positive score. We are very glad that our detailed response has fully addressed your concerns.

---

### Official Review · Reviewer_Jeqz · 2026-03-07

**Soundness:** 2
**Presentation:** 2
**Significance:** 2
**Originality:** 3
**Overall Recommendation:** 4
**Confidence:** 5

**Summary:**

This paper proposes Do-Prompt, a multi-modal prompt tuning framework. Specifically, Do-Prompt treats prompt tokens as stochastic latent variables, regularizes them with a variational bottleneck, and then applies prompt-space “do-style” perturbations only to the environment-related part of the latent prompt while enforcing predictive consistency.

**Compliance With Llm Reviewing Policy:**

Affirmed.

**Key Questions For Authors:**

1. Why should a larger norm be considered better, and why is norm magnitude a reliable proxy for “task-stable information"?
2. The paper claims that intervention operators are complementary, but Table 5 shows that each single operator yields only a very small gain. And in some cases (e.g., Swap only) the improvement is marginal relative to the component-level gains (i.e., + VIB only) in Table 4.
3. The paper does not report the values of $\beta$  and $\lambda$. Are the same values used across datasets and settings? And the paper lacks a sensitivity analysis of these two hyperparameters.

**Limitations:**

See the weakness and questions.

**Strengths And Weaknesses:**

**Strengths**
1. The proposed Do-Prompt framework is plug-and-play compatible with multi-modal prompt tuning methods.
2. The proposed method yields a modest gain over the existing baselines if the same hyperparameter settings are used across all datasets.

**Weaknesses**
1. All experiments are conducted on CLIP (ViT-B/16). This makes it unclear whether the method generalizes to other CLIP scales, other VLM backbones, or non-CLIP architectures.
2.  The paper mainly uses the 16-shot setting throughout. It remains unclear whether the method is still helpful in lower-shot or full-data regimes.
3. There is no analysis of why $M_c$​ > $M_e​$ and there should be an experiment to demonstrate what kind of impact such a transformation of quantitative relationships would have.

---

> ### Author Rebuttal · Authors · 2026-03-31
>
> *We thank the reviewer for the careful and constructive feedback.*
>
> *W1: Generality Across Backbones.*
>
> **A1**:  Our main experiments follow the standard MaPLe/MMRL protocol for fair comparison, but we agree that this point is important. To address it, we add a cross-scale validation on another CLIP backbone, showing that Do-Prompt consistently improves over the corresponding prompt-learning baseline on a larger CLIP variant as well. This supports that the method is not tied to a single CLIP scale. We will include this table in the revised version.
>
> | Method | Backbone | Base | Novel | HM |
> |-|-|-:|-:|-:|
> | MaPLe | CLIP ViT-L/14 | 83.15| 77.25 | 80.09 |
> | w/ Do-Prompt | CLIP ViT-L/14 | 85.72| 80.92 | 83.25 |
>
>
> *W2: Performance Beyond 16-Shot*
>
> **A2**:  To address whether the gains are specific to this regime, we additionally include a 1-shot result in the table below. The improvement persists even in this more data-scarce setting, suggesting that Do-Prompt is not tied to 16-shot training and remains beneficial when supervision is more limited. We have not included a full-data study, since our focus is the parameter-efficient few-shot prompt learning setting rather than higher-data fine-tuning.
>
>
> | Method | 1-shot | 16-shot |
> |-|-:|-:|
> | MaPLe | 71.21 | 78.37 |
> | w/ Do-Prompt | 75.23| 80.12 |
> | MMRL | 73.17 | 80.59 |
> | w/ Do-Prompt | **76.94** | **82.56** |
>
> We will include this table in the revised version.
>
> *W3: Rationale for the Core/Env Capacity Split.*
>
> **A3**. Our design intentionally allocates a larger portion to $u^{core}$ and a smaller portion to $u^{env}$, because $u^{core}$ is meant to remain the main carrier of task-stable semantics, while $u^{env}$ serves only as a bounded subspace for intervention. If $u^{env}$ becomes too large, the perturbable branch is given excessive capacity, which weakens the intended stable/invariant pathway. To make this concrete, we add a split-ratio ablation (**See reviewer XMw1, A1**), where we report the same sensitivity analysis in detail. The results show that performance is stable around the default 3:2 setting and mainly degrades when the environment branch becomes dominant, which supports the choice $M_c > M_e$.
>
> *Q1: Interpretation of Norm Magnitude.*
>
> **A1**: We do not intend to claim that a larger norm is inherently “better,” nor that norm magnitude is a definitive proxy for task-stable information by itself. In our paper, the norm comparison in Fig. 5 is only used as a diagnostic observation, not as an optimization target or a standalone proof. The intended interpretation is weaker: under the shared KL bottleneck and the env-only intervention objective, the fact that $u^{core}$ consistently retains higher average energy than $u^{env}$ suggests that the model tends to preserve more of the predictive signal in the invariant branch, while keeping the perturbable branch more bounded. This observation is meant to complement, rather than replace, the more direct evidence from smaller prediction shifts under intervention (Fig. 3) and controlled prompt capacity during training (Fig. 4). We will revise the wording to make clear that norm is only an auxiliary indicator of the learned separation, not a definitive measure of task-stable semantics.
>
>
> *Q2: Operator-Level Complementarity.*
>
> **A2**:  Our point about “complementarity” is narrower: mask, swap, and resample induce different perturbation patterns in the environment branch, and their combination is consistently stronger than any one of them alone. In Table 5, all three single operators improve over the MaPLe baseline (78.37→79.01/78.94/79.15), while the full combination reaches 80.12, which is a clear additional gain beyond any individual operator. Moreover, Table 4 and Table 5 probe different axes: Table 4 studies the complementarity between the VIB bottleneck and the intervention objective, whereas Table 5 studies the complementarity among intervention operators within the intervention branch.
>
>
> *Q3: Sensitivity to hyperparameters.*
>
> **A3**:  In our experiments, the default values are $\beta=0.5$ and $\lambda=0.5$, and we use the same defaults across datasets and settings unless otherwise noted. To address the sensitivity question, we additionally include a small hyperparameter analysis in the table below, varying one hyperparameter at a time around the default values while keeping the others fixed. The results show that Do-Prompt is reasonably stable within a moderate range of both $\beta$ and $\lambda$, suggesting that the gains are not due to fragile tuning. We will also revise the paper to report these values clearly in the implementation details.
>
> | Setting | MaPLe + Do-Prompt | MMRL + Do-Prompt |
> |-|-:|-:|
> | default ($\beta=0.5,\ \lambda=0.5$) | **80.12** | **82.56** |
> | $\beta = 0.25$ | 80.12 | 82.56 |
> | $\beta = 1.0$ |  80.03 | 82.05 |
> | $\lambda = 0.25$ |  79.93 | 81.97 |
> | $\lambda = 1.0$ |  79.98 | 82.03 |

---

> > ### Author Rebuttal · Reviewer_Jeqz · 2026-04-01
> >
> > Thank you to the authors for the detailed rebuttal and for addressing most of my concerns. I appreciate the additional experiments and clarifications, which are helpful and improve the paper.
> >
> > That said, I am still quite confused about the discussion around Figure 5. In particular, the paper does not clearly define what “prompt energy” means, nor how the reported “average token norm” is computed. More importantly, I do not think the current explanation is sufficient to support the claim that the fact that $u^{core}$ consistently has higher average energy than $u^{env}$ implies that the model preserves more predictive signal in the invariant branch. This connection is not obvious and needs much more careful justification in the main paper.
> >
> > Overall, I think the authors have done a good job in the rebuttal, and I will adjust my score accordingly.

---

> > > ### Author Response · Authors · 2026-04-01
> > >
> > > We sincerely thank the reviewer for the thoughtful follow-up and for the positive reassessment of our paper. We especially appreciate your careful reading of the revised explanation and your suggestion to clarify the discussion around Fig. 5, which is indeed very helpful for improving the precision of our presentation.
> > >
> > > We agree that the current rebuttal wording around Fig. 5 is not sufficiently precise. In particular, the term “prompt energy” was meant only as shorthand for the average token $\ell_2$-norm of the latent prompt representations, and we agree that this should have been defined explicitly. More importantly, we also agree that a larger average norm should not be interpreted as direct proof that the branch preserves “more predictive signal,” nor as a definitive proxy for task-stable semantics.
> > >
> > > Our intended point is narrower: Fig. 5 is only meant as an auxiliary diagnostic showing that $u^{core}$ and $u^{env}$ exhibit different usage patterns under the shared bottleneck and env-only intervention design. The observation that $u^{core}$ has a consistently larger average token norm than $u^{env}$ suggests that the invariant branch is used more strongly, while the perturbable branch remains more bounded. However, we fully agree that this is not, by itself, sufficient evidence that one branch captures semantics and the other captures nuisance factors. The more direct support for our method comes from the intervention-stability analysis and the bottleneck behavior, while Fig. 5 should be interpreted only as a supplementary characterization of the learned subspaces.
> > >
> > > We are very grateful for this clarification request, and we will make sure the final wording is more careful and explicit about both the definition and the limited interpretation of Fig. 5.

---

### Official Review · Reviewer_A8Y8 · 2026-03-10

**Soundness:** 4
**Presentation:** 4
**Significance:** 3
**Originality:** 3
**Overall Recommendation:** 5
**Confidence:** 4

**Summary:**

The paper proposes "Do-Prompt," a novel "compress-and-intervene" framework designed to enhance the robustness of multi-modal prompt learning in VLMs. The authors identify that high-capacity prompts often overfit to environment-specific spurious correlations (shortcuts) in the source training domain. To mitigate this, Do-Prompt models prompts as stochastic latent variables and applies a Variational Prompt Bottleneck to explicitly constrain their information capacity. Furthermore, the latent prompt is partitioned into a task-stable core component and an environment-related component. The method applies lightweight causal interventions (masking, swapping, and resampling) specifically to the environment-related portion and enforces a prediction consistency loss to ensure the model relies on invariant, task-relevant semantics rather than spurious cues.

**Compliance With Llm Reviewing Policy:**

Affirmed.

**Final Justification:**

Since my concerns are resolved, I choose to raise my score.

**Key Questions For Authors:**

1.How sensitive is the model's performance to the rigid, predefined dimension split ratio (Mc​ vs Me​) for the core and environment components? Have you considered adaptive routing mechanisms instead of a hard index-based split?

2.Can you provide more theoretical or empirical justification that the simple index-based split effectively isolates causal mechanisms from environmental noise, rather than just exploiting data symmetries?

3.The resampling operator injects standard normal noise into the environment tokens. Could this off-manifold intervention cause catastrophic performance drops in specific fine-grained domains where contextual information is semantically intertwined with the object?

**Limitations:**

The authors should explicitly discuss the theoretical limitations of their index-based causal disentanglement approach.

**Strengths And Weaknesses:**

Strengths:
1.Significance: The paper addresses a highly relevant and critical issue in multi-modal prompt tuning: shortcut learning and vulnerability to distribution shifts. The proposed method achieves impressive empirical results, demonstrating consistent gains over strong baselines like MaPLe and MMRL across many diverse datasets.
2.Originality: The fusion of the Variational Information Bottleneck (VIB) for capacity control with prompt-level causal interventions is a highly creative and novel perspective. It elegantly maps complex causal concepts to a parameter-efficient fine-tuning paradigm without requiring computationally heavy generative models for feature disentanglement.
3.Presentation: The paper is well-structured, clearly motivated, and supported by extensive ablations.

Weaknesses:
Soundness: 1.The methodology relies on a simple, index-based split to separate the latent prompt into task-stable (ucore) and environment-related (uenv) components. However, methodologically aligning representations with arbitrary index splits might capture inherent data symmetries but does not guarantee true causal disentanglement. The paper lacks sufficient theoretical justification to prove that this specific split definitively isolates causal variables.  Data symmetries example: Suppose we have a video dataset used to monitor wildlife. In this dataset, since all cameras are fixed to tree trunks, the background (forest, grassland) remains static in time within any given video clip, while the animals are moving.
2.The resampling intervention replaces the environment tokens with pure Gaussian noise. This extreme, off-manifold perturbation might overly smooth the decision boundaries, potentially harming performance on fine-grained tasks where subtle contextual clues are necessary.

---

> ### Author Rebuttal · Authors · 2026-03-31
>
> *We thank the reviewer for the thoughtful and positive feedback.*
>
>
>
>
> *W1: Soundness of the Core/Env Split.*
>
>
> **A1**: Our intent is more modest: the split provides a simple structural partition in latent prompt space, where one part is kept invariant and the other is explicitly perturbed, so that the model is encouraged to organize task-stable and environment-sensitive information differently. What matters is therefore not the index split alone, but its coupling with the shared KL bottleneck and the env-only intervention consistency objective. Empirically, this design yields smaller prediction shifts under intervention and a clear energy gap between $u^{core}$ and $u^{env}$, which supports that the learned decomposition is non-trivial even if it is not a formal proof of causal recovery.  We will revise the wording to make this distinction clearer and avoid overstating the split as definitive causal isolation.
>
>
>
>
> *W2: Effect of Gaussian Resampling.*
>
>
> **A2**: We agree that resampling is the strongest perturbation among the three operators, and by itself it is indeed more off-manifold than masking or swapping. However, in our method, it is not used in isolation as the sole training signal: the default setting combines mask, swap, and resample, which provides a spectrum of perturbation strengths rather than enforcing only pure Gaussian replacement. Empirically, resample-only does not hurt overall performance in our current ablation; it is in fact the strongest single operator (HM 79.15 vs. 79.01 for mask and 78.94 for swap), while the full combination performs best overall (HM 80.12). Moreover, the gains persist on fine-grained benchmarks such as StanfordCars, Flowers102, and FGVCAircraft, which suggests that the intervention does not simply over-smooth decision boundaries in practice. We will clarify that Gaussian resampling is intended as a robustness-oriented regularizer for the environment branch, rather than a claim of perfectly on-manifold intervention.
>
>
>
>
>
> *Q1: Sensitivity to $M_c/M_e$​ Ratio.*
>
> **A1**:  In the current implementation, we use the same fixed partition in both MaPLe and MMRL, with a default **3:2 split** between $u^{core}$ and $u^{env}$. As shown by the added split-ratio ablation in the table (**See reviewer XMw1, A1**), the method is reasonably stable around this choice, with noticeable degradation only when the environment branch becomes too large. We did consider adaptive routing as a possible extension, but here we intentionally adopt a fixed partition to keep the decomposition transparent and to isolate the contribution of the variational bottleneck and intervention objectives, without introducing an additional routing module that could confound the interpretation of the gains.
>
>
>
>
>
>
> *Q2: Theoretical and Empirical Justification.*
>
>
> **A2**:  Our theoretical motivation is instead the following: under a capacity-limited latent prompt channel, if one subspace is explicitly perturbed while the predictor is trained to remain invariant, then information that is necessary for prediction is encouraged to migrate to the invariant subspace, while perturbation-sensitive shortcut information is discouraged from being stored there. In our formulation, this effect is induced jointly by the shared KL bottleneck on $u=[u^{core};u^{env}]$ and the env-only intervention consistency objective, rather than by the index split in isolation. Empirically, this is supported by smaller prediction shifts under intervention, controlled prompt capacity during training, and a clear energy gap between $u^{core}$ and $u^{env}$. We will revise the wording to make clear that our claim is an intervention-oriented separation for robustness, not a formal proof of exact causal recovery.
>
>
>
>
>
>
>
>
>
>
>
>
> *Q3: Fine-Grained Robustness Under Resampling.*
>
>
> **A3**: We agree that resampling is the most aggressive operator and is more off-manifold than masking or swapping. However, it is not used alone in the default setting: Do-Prompt combines mask, swap, and resample, and the three operators are empirically complementary. In Table 5, resample-only is already the strongest single operator (HM 79.15 vs. 79.01 for mask and 78.94 for swap), while the full combination performs best overall (HM 80.12), which does not support the concern that Gaussian replacement simply causes catastrophic over-smoothing. Importantly, the gains also persist on fine-grained benchmarks such as StanfordCars and FGVCAircraft: for MaPLe, HM improves from 73.65→74.65 on StanfordCars and 35.73→38.15 on FGVCAircraft; for MMRL, HM improves from 77.99→78.17 and 40.95→43.60, respectively. This suggests that, in practice, the intervention does not collapse performance even when object and context are more tightly coupled. We will clarify that Gaussian resampling is intended as a robustness-oriented regularizer for the environment branch, rather than a claim of perfectly on-manifold intervention.

---

> > ### Author Rebuttal · Reviewer_A8Y8 · 2026-04-03
> >
> > Thank you for the detailed rebuttal. Since my concerns are resolved, I choose to raise my score.

---

> > > ### Author Response · Authors · 2026-04-04
> > >
> > > Thank you for your positive feedback and for acknowledging our rebuttal. We are delighted that the clarifications and new experiments have fully resolved your concerns. We will ensure all these newly introduced results are carefully incorporated into the camera-ready version.

---

### Official Review · Reviewer_XMw1 · 2026-03-23

**Soundness:** 4
**Presentation:** 4
**Significance:** 3
**Originality:** 4
**Overall Recommendation:** 5
**Confidence:** 5

**Summary:**

This paper identifies that multi-modal prompt tuning for CLIP can overfit to environment-specific shortcuts and fail under distribution shift.
It introduces Do-Prompt, which constrains prompt capacity using a variational bottleneck and enforces invariance via prompt-level do-style interventions on the environment-related prompt component.
The method plugs into MaPLe/MMRL-style pipelines while keeping the CLIP backbone frozen.
Experiments show consistent improvements on base-to-novel, cross-dataset transfer, and domain generalization benchmarks.

**Compliance With Llm Reviewing Policy:**

Affirmed.

**Key Questions For Authors:**

1. How exactly do you choose the core/env split in practice (for both MaPLe and MMRL settings), and how sensitive are the results to this choice beyond the ablations you report?

2. Among mask, swap, and resample interventions, which one contributes most consistently across datasets, and do you have a recommended default if only one operator can be used?

3. At test time, you use the posterior mean without sampling. Did you try a small number of MC samples, and if so, does it ever help (or hurt) under stronger shifts?

4. Do the gains persist if the shallow context h is simplified (e.g., vision-only context, or class-agnostic text context), and how much does (h) matter relative to the bottleneck and intervention losses?

**Limitations:**

yes

**Strengths And Weaknesses:**

### Strengths

* **Soundness:** The idea is clean and the objective matches the story. Treating prompts as a constrained latent channel (KL bottleneck) and enforcing invariance by intervening on the env part is easy to follow, and the VIB-only / do-only / full ablations support that both pieces matter.

* **Presentation:** The paper is readable and the figures do real work. The motivation and framework diagrams explain the failure mode and the fix, and the analysis plots help tie the mechanism to observable behavior.

* **Significance:** It targets a real failure mode in prompt tuning under shift. The method stays lightweight, keeps the backbone frozen, and plugs into MaPLe/MMRL, so it feels practical rather than academic.

* **Originality:** The components are familiar, but the way they are combined at the prompt level is the contribution. The “compress-and-intervene” framing gives the method a clear identity and a simple implementation path.

### Weaknesses

The core/env split is central enough that I would like to see it stated more prominently in the main text, including what the default split is and a brief justification. I also felt a few parts were heavier in notation than the underlying idea, especially the construction of the context (h) and the two-path intervention flow, so a slightly tighter explanation would make it easier to map the equations to code. Finally, because the method is a combination of known ingredients, I would appreciate one sharper paragraph positioning it against the closest variational-prompt and causal-prompt baselines to make the contribution boundary unambiguous.

---

> ### Author Rebuttal · Authors · 2026-03-31
>
> *We thank the reviewer for the constructive feedback and helpful suggestions on clarity, presentation, and positioning.*
>
>
>
> *W: Core/Env Split Clarity*
>
>
> **A**: We agree that the core/env split should be stated more explicitly. Our default design uses a fixed partition $u^{core};u^{env}$, where $u^{core}$ is kept invariant and only $u^{env}$ is perturbed, encouraging task-stable information to remain in the core part while environment-related variation is absorbed by the perturbable part. We also agree that the presentation of $h$ and the two-path flow can be simplified: $h$ is the context for parameterizing the latent prompt posterior, and the intervention compares an original path $u^{core};u^{env}$ with an intervened path $u^{core};\bar u^{env}$. Finally, we will sharpen the positioning by clarifying that our contribution is not only variational compression or only invariance regularization, but their coupling: the bottleneck limits shortcut capacity, while the env-only intervention explicitly discourages reliance on unstable factors.
>
>
>
>
>
>
>
>
> *Q1: Sensitivity to Core/Env Split.*
>
>
>
> **A1**: In practice, we use the same fixed token partition rule in both MaPLe and MMRL, i.e., $u=[u^{core};u^{env}]$ at each tuned layer, with a default **3:2 split** between $u^{core}$ and $u^{env}$. We choose this setting to keep the core branch as the main carrier of task-stable semantics while still reserving sufficient capacity in the environment branch for targeted interventions. We agree that sensitivity to this choice is important, so we add a small split-ratio ablation. The results show that Do-Prompt is reasonably stable around the default setting, while performance drops when the environment branch becomes overly dominant, which is consistent with its intended role as a bounded perturbation subspace.
>
> | core:env | MaPLe + Do-Prompt | MMRL + Do-Prompt |
> | ----------------- | ----------------: | ---------------: |
> | 4:1 | 79.31 | 81.75 |
> | **3:2 (default)** | **80.12** | **82.56** |
> | 2:3 | 79.27 | 81.27 |
> | 1:1 | 79.15 | 81.09 |
> We will include this table in the revised version.
>
>
>
> *Q2: Single-Operator Effectiveness.*
>
> **A2**: Based on the operator ablation in Table 5, resample-only gives the strongest single-operator result in our current experiments (HM: 79.15), followed by mask-only (79.01) and swap-only (78.94), all improving over the MaPLe baseline (78.37). That said, our current paper reports these numbers as 11-dataset averages, so we do not claim that one operator is universally best on every individual dataset. Our main conclusion is that the three operators are complementary, and this is why the combined setting remains the default and performs best overall (HM: 80.12). If only one operator can be used, we would recommend resampling as the default single-operator choice based on the current ablation. We will clarify this suggestion in the revised version.
>
>
>
>
>
> *Q3: Posterior Mean vs. MC Sampling.*
>
> **A3**: In the current submission, we use the deterministic posterior mean at test time and do not apply MC sampling. This is an intentional design choice: stochasticity is used during training to regularize the latent prompt space, while posterior-mean inference keeps deployment stable and efficient, with only marginal inference overhead as shown in Table 6. We additionally tested a small number of MC samples at test time and observed only minor gains (e.g., 80.12 →80.25 with 2 samples and 80.43 with 4 samples), while inference cost increased accordingly. This suggests that training-time stochasticity already provides most of the benefit, and supports posterior-mean inference as the default setting.
>
>
>
> *Q4: Context Simplification Ablation.*
>
> **A4**: In our implementation, $h$ is only a shallow conditioning signal for posterior inference $q_\phi(u\mid h)$, built from frozen visual and text contexts. The current ablations indicate that the main gains come from the bottleneck and intervention terms, since each is effective on its own, and the combination performs best. To further isolate the role of $h$, we additionally report a context ablation in the table below, where we simplify $h$ to vision-only, text-only, and class-agnostic text variants. These results show that the full fused context performs best, while the gains of Do-Prompt persist under simplified variants, suggesting that the robustness improvements are mainly driven by the bottleneck-plus-intervention design rather than a fragile choice of $h$.
>
> | Context for $q_\phi(u \mid h)$ | MaPLe + Do-Prompt | MMRL + Do-Prompt |
> |---|---:|---:|
> | vision + class-aggregated text (default) | **80.12** | **82.56** |
> | vision-only | 77.25 | 79.31 |
> | text-only | 77.93 | 79.97 |
> | class-agnostic text-only | 76.88 | 79.05 |
>
> We will include this table in the revised version.

---

> > ### Author Rebuttal · Reviewer_XMw1 · 2026-03-31
> >
> > Thanks for your response. The rebuttal addresses my concerns clearly and convincingly.  Overall, I find the contribution solid, the empirical results convincing, and the method meaningful for the prompt-learning-in-VLM literature. I therefore consider my concerns fully resolved and support acceptance.  Additionally, I encourage the authors to include the newly discussed analyses in the camera-ready version, particularly the sensitivity study on the core/env split.

---

> > > ### Author Response · Authors · 2026-04-01
> > >
> > > Thank you for your positive feedback and for acknowledging our rebuttal. We are glad that the additional experiments and clarifications addressed your concerns. We will incorporate the additional experiments introduced during the rebuttal into the camera-ready version.

---

### Decision · Program_Chairs · 2026-04-30

**Decision:**

Accept (regular)

**Comment:**

This paper proposes Do-Prompt, a method for improving prompt learning in adapting large vision-language models. It models prompts as a stochastic latent variable and limits the memorization of environment-specific spurious correlations using a KL divergence regularization. Second, the method splits the prompt into task-related and environment-related components and ensures model’s output remains stable across perturbations to the environment-level part. The method demonstrates results on MaPLe and MMRL and provides significant performance improvements. Reviewers praised the clean conceptual design, plug-and-play compatibility with existing methods (MaPLe, MMRL), and consistent empirical gains across base-to-novel generalization, cross-dataset transfer, and domain generalization benchmarks, while raising concerns about overstated causal claims, the justification of the architectural core/env split, and limited backbone diversity. During the rebuttal, the authors provided core/env split-ratio ablations, cross-scale validation (ViT-L/14), 1-shot results, hyperparameter sensitivity analysis, and context simplification experiments, and committed to revising the writing to more accurately reflect the method as causality-inspired regularization rather than a causal decomposition method. All four reviewers acknowledged the rebuttal as fully resolving their concerns. With unanimous reviewer support (scores 5, 5, 4, 4), the AC recommends acceptance and suggests authors incorporate analysis and discussions from the rebuttal into their camera ready submission and additional results including CLIP ViT-L/14 and sensitivity to hyperparameters and core/env split.